# Anti-Fouling and Anti-Biofilm Performance of Self-Polishing Waterborne Polyurethane with Gemini Quaternary Ammonium Salts

**DOI:** 10.3390/polym15020317

**Published:** 2023-01-07

**Authors:** Yi Zhang, Tao Ge, Yifan Li, Jinlin Lu, Hao Du, Ling Yan, Hong Tan, Jiehua Li, Yansheng Yin

**Affiliations:** 1Engineering Technology Research Center for Corrosion Control and Protection of Materials in Extreme Marine Environment, Guangzhou Maritime University, Guangzhou 510725, China; 2College of Polymer Science and Engineering, State Key Laboratory of Polymer Materials Engineering, Sichuan University, Chengdu 610065, China; 3State Key Laboratory of Metal Material for Marine Equipment and Application, Anshan 114000, China

**Keywords:** anti-fouling, anti-biofilm, self-polishing, waterborne polyurethane

## Abstract

Biofilms are known to be difficult to eradicate and control, complicating human infections and marine biofouling. In this study, self-polishing and anti-fouling waterborne polyurethane coatings synthesized from gemini quaternary ammonium salts (GQAS), polyethylene glycol (PEG), and polycaprolactone diol (PCL) demonstrate excellent antibiofilm efficacy. Their anti-fouling and anti-biofilm performance was confirmed by a culture-based method in broth media, with the biofilm formation factor against Gram-positive (*S. aureus*) and Gram-negative bacterial strains (*E. coli*) for 2 days. The results indicate that polyurethane coatings have excellent anti-biofilm activity when the content of GQAS reached 8.5 wt% against *S. aureus*, and 15.8 wt% against *E. coli*. The resulting waterborne polyurethane coatings demonstrate both hydrolytic and enzymatic degradation, and the surface erosion enzymatic degradation mechanism enables them with good self-polishing capability. The extracts cyto-toxicity of these polyurethane coatings and degradation liquids was also systematically studied; they could be degraded to non-toxic or low toxic compositions. This study shows the possibility to achieve potent self-polishing and anti-biofilm efficacy by integrating antibacterial GQAS, PEG, and PCL into waterborne polyurethane coatings.

## 1. Introduction

Microorganisms tend to adhere and aggregate on the surfaces of materials; then, through the interaction between microbes and materials and between microbes themselves, they gradually adhere and fix on the surface of materials, further grow and reproduce, and form biofilms. These biofilms can affect many areas, including public health, food processing, transportation, and marine industrial equipment [1]. Biofilms are known to complicate the damage by protecting bacteria through reducing antibacterial agents’ efficacy, and dispersing planktonic bacteria to distant sites to start a new biofilm. These biofilms might change the physico-chemical property of material surface environment and lead to local or pitting corrosion [2], which significantly affects the service behavior, service time, and maintenance cost of engineering facilities. Besides, in the process of cleaning the marine fouling, biological invasion and epidemic spread may also happen and lead to economic and social harm [3]. Since biofilm formation on equipment has been acknowledged as the main problem of microbial contamination, surface modification seems to be of great urgency. To minimize the damage to the bulk properties of equipment and reduce costs, anti-fouling coatings with effective microbicidal or anti-adhesion performance has been considered as an attractive approach to fight microbial contamination.

Waterborne polyurethane has been widely used in biomedical areas, wood coatings, leather coatings, architectural coatings, corrosion protective coatings, and other fields [4,5]. Polyurethane is a multi-block polymer with good molecule tailing capability, which can quantitatively introduce various functional groups into the molecular chain through simple molecular structure design [6,7]. For example, introducing quaternary ammonium salt into the molecular chain of polyurethane can give killing property of polyurethane, and silicon and fluorine chain can reduce the coating surface energy and releasing contaminants [8,9,10]. However, coatings with only biocidal components requires a large dosage to achieve anti-fouling effect, and the dead microorganisms will adhere to the coatings’ surface, which can hinder the further biocidal activity and lead to the formation of biofilm [11,12]. Coatings with amphiphilic components, such as zwitterionic compounds and polysaccharides [13], poly(ethylene glycol) (PEG) [14], and gels [15] have showed extremely good fouling-resistant properties, which are widely used for fouling-resistant coatings. Silicone- or fluorine-based polymers [16] with hydrophobicity, low surface energy, and low elastic modulus have the best foul-release performance, which is considered as foul-release coatings. However, the low surface energy and low elastic modulus accompanied by the hydrophobicity could lead to less interaction of the coated surface with other materials and reduce the adhesive force of the coating [17]. What is more, fouling-resistant or foul-release coatings cannot completely preclude the adhesion of microorganisms, and once adhesion occurs, microorganisms will multiply and eventually lead to the loss of anti-fouling function [18,19]. Therefore, coatings having the ability to resist fouling adhesion and kill fouling organisms are potential anti-fouling measures [20,21,22].

Chen et al. prepared urushiol-based benzoxazine copper polymer coatings containing natural product urushiol, paraformaldehyde, and CuCl_2_, which exhibited excellent anti-fouling performance [23]. Coatings coupling biocidal agents are known to have good environmental stability and biological activity, extended residence time, and low toxicity compared with releasing biocidal agents, which makes them suitable for use in the field of marine anti-fouling [24]. Therefore, Yang et al. prepared coatings that integrated both antibacterial and fouling-resistant properties by surface grafting PEG-b-cationic polycarbonate block copolymers [25]. Tiller et al. reported a surface modification of silicone with 3-(trimethoxysilyl)-propyldimethyloctadecyl ammonium chloride, which resulted in a surface with both Lotus-Effect and contact-active antimicrobial properties [26]. Jiang et al. prepared a switchable biocompatible surface through grafting poly (N, N-dimethyl-N-(ethoxycarbonylmethyl)- N-[2-(methacryloyloxy)- ethyl] -ammonium bromide). This surface could kill bacteria on the one hand, and then switch to a zwitterionic anti-fouling surface, releasing dead bacteria by hydrolysis [12]. Zhou et al. synthesized sol-gel-derived hard coatings containing trimethoxysilyl and isothiazolinone groups [15]. Liu et al. prepared self-healing polydimethylsiloxane anti-fouling surface through zwitterionic polyethyleneimine-functionalized gallium nanodroplets [16]. Vamvakaki et al. prepared biocompatible antimicrobial hybrid coatings combining enhanced biocidal activity and stimuli-renewable antimicrobial action based on a natural polymer, chitosan, modified to bear QAS groups, and modified TiO_2_ nanoparticles [27].

These coatings introducing antibacterial and anti-adhesion groups could give them good anti-fouling properties, and cationic ammonium salt is one of the most-used antibacterial components, such as quaternary ammonium salt [28]. Nevertheless, gemini quaternary ammonium salt (GQAS) have much stronger biocidal activity than traditional single-chain QAS, which are known to have good environmental stability and biological activity, extended residence time, low toxicity to mammalian cells [29,30]. Therefore, our group has developed anti-fouling waterborne polyurethane emulsions by introducing GQAS chain extenders and PEG polydiol, which would be promisingly applied in biomedical areas [31,32,33]. These works demonstrated that GQAS with 8–12C hydrophobic alkyl tails inside chains can efficiently kill these bacteria on contact. However, there is no evidence to prove these coatings contain antibacterial groups and anti-adhesive groups can certainly break biofilms or prevent biofilm formation.

Self-polishing antifouling coating could also be a choice in antifouling applications [34]. Zhou et al. developed a series of interpenetrating polymer networks (IPNs) equipped with zwitterionic nanoparticles and lubricant for quasistatic and dynamic antifouling [35]. Zhou et al. prepared a self-polishing antifouling coating via a borneol/boron acrylate polymers, which can remove the adhered biofouling by hydrolysis to renew the surface [36]. Zhang et al. developed coatings with dynamic anti-fouling surfaces by using degradable polyester-based polymers, which having tunable renewability through gradual degradation [37,38,39,40]. Therefore, a continuously changing dynamic anti-fouling surface can effectively prevent fouling, including bacteria and other marine organisms, from landing, attachment, and proliferation. Thus, degradability of anti-fouling coatings was also an important factor for designing anti-fouling coatings under complicated conditions in microbial contamination, public health, or marine environments.

Herein, a series of anti-fouling waterborne polyurethane emulsions were synthesized using isophorone diisocyanate (IPDI), PEG and poly(e-caprolactone) diols (PCL) as raw materials, and L-lysine and GQAS diamine (EG12) were used as chain extenders (Figure 1). These obtained waterborne polyurethanes are named as WPUn, where n represented the solid content of EG12 in polyurethane. The effects of EG12 contents on the bulk structure of the obtained waterborne polyurethanes were examined by gel permeation chromatography (GPC), proton nuclear magnetic resonance (^1^HNMR), differential scanning calorimetry (DSC), and Fourier transform infrared spectroscopy (FTIR). Both hydrolytic and enzymatic degradation experiments were systematically processed to study their self-polishing ability. The anti-fouling and biofilm resistance activity of these WPUn coatings were evaluated via protein adsorption assay, contact-active killing assay, and culture-based assay with biofilm formation factor, against *E. coli* (Gram-negative) and *S. aureus* (Gram-positive). Cytotoxicity analysis was also performed to study the effect of EG12 on obtained WPUn samples. Such biodegradable self-polishing coatings could potentially be used in surgical equipment, health care, and marine anti-fouling areas.

## 2. Materials and Methods

### 2.1. Materials

All materials were obtained from commercial suppliers and used directly unless mentioned otherwise. Chain extender EG12 (12 is the number of carbon atoms in the hydrophobic alkyl chain) was synthesized in our previous work [31], and the synthetic route is illustrated in Appendix A. Lysine was purchased from Aladdin reagent (Shanghai, China). Glutaraldehyde, MH broth, LB broth, TSB broth, acid hydrolyzed casein, and glucose were purchased from Qingdao Haibo Biotechnology limited company (Qingdao, China). Lipase from Pseudomonas cepacia (lipase PS, ≥30 U/mg) was obtained from Sigma Aldrich (Shanghai, China). PEG (molecular weight 1450) and PCL (molecular weight 2000) were purchased from Dow Chemical (Midland, MI, USA). IPDI obtained from BASF (Ludwigshafen, German) was redistilled under vacuum before use. Thermoplastic polyether polyurethane (TPEU) was obtained from Dongguan City grindlays film Products Company (Dongguan, China).

### 2.2. Preparation WPUn Emulsions and Coatings

A series of self-polishing WPUn emulsions based on IPDI, PCL, PEG, lysine, and EG12 were synthesized by two-step polymerization. The feed ratios were shown in Table 1. The content of GQAS in WPUn structure was regulated by changing the ratio of two chain extenders. PEG and PCL were dehydrated under vacuum at 80–90 °C for 2 h first. Then, IPDI and 0.1% stannous octoate were added under N_2_ environment. After continuous reaction for another 2 h at 70–75 °C, the reaction prepolymer was cooled down. Then, different content of chain extender EG12 in acetone was added in the mixture for 0.5 h. Afterward, the polyurethane prepolymer was poured into L-lysine aqueous solution under stirring (600 rpm) and ultrasound (200 W) simultaneously for 1 h. An amount of 1 M NaOH was added dropwise to neutralize L-lysine and accelerate the secondary chain extension. WPU0 without GQAS was prepared as a control with a similar process.

WPUn coatings were prepared with a similar process to our previous work [32] by casting these WPUn emulsions siliconized culture dishes and dried at room temperature for 2 days, 60 °C for 2 days, and then 60 °C under vacuum for 2 days. The films were cut in sheets (1 × 1 cm^2^, 0.5 mm thickness) and used directly for tests unless mentioned otherwise. To eliminate the influences of unreacted PEG and GQAS moieties onto WPUn films of their anti-fouling activities, all the films were immersed in water (110 rpm, 37 °C) for 6 h, then put into an oven at 60 °C for 2 days before testing, including antibacterial, anti-biofilm activity, protein adsorption assay, and cell cytotoxicity.

### 2.3. Degradation Assay

Hydrolytic degradation medium (PBS, pH = 7.4) and enzymatic degradation medium (PBS + lipase PS, pH = 7.4, lipase PS concentration: 0.3 mg·mL^−1^) were prepared before the test. WPUn films (1 × 1 cm^2^, 0.5 mm thickness) were weighed (*W*_0_) and immersed in a glass tube containing 2 mL degradation medium, then the vials were incubated with shaking (110 rpm) at 37 °C [41,42]. Periodically removed samples were washed with distilled water three times, dried in vacuum oven at 25 °C to a constant weight (*W_t_*). The degradation medium was changed every 2 days with fresh medium. The weight loss was calculated as the following:$$Weight\;loss\;\left(\%\right)=\frac{W_{0}-W_{t}}{W_{0}}\times 100\;$$

The morphology of WPUn after degradation was characterized by a scanning electron microscope microscopy (SEM).

### 2.4. Anti-Fouling Behavior

MIC studies: The MIC of WPUn emulsions was conducted by method according to the national committee for clinical laboratory (NCCLS) against *E. coli* (ATCC 25922) and *S. aureus* (ATCC 6538). Bacteria cultures in the nutrient media MH were diluted to 10^6^–10^7^ cells·mL^−1^ before the test. EG12 and DTAB (1024 mg·mL^−1^) were used as controls. Wells containing only media were used as negative control, while only diluted bacteria culture were set as positive control. After incubation at 37 °C, 110 rpm for 16–18 h, 5 μL triphenyltetrazolium chloride (5 mg·mL^−1^) was added into each well for 0.5 h before the results were read.

Protein adsorption: The protein adsorption of these WPUn films was tested by bull serum albumin (BSA, dissolved in the isotonic PBS, pH = 7.4). TPEU (0.5 mm thickness) was selected as a control. WPUn and TPEU films were equilibrated in PBS overnight, and then incubated in BSA solution (1 mg·mL^−1^) at 37 °C for 2 h, and gently rinsed with sterile PBS and deionized water. The adsorbing protein in these samples was eluted in the SDS solution and quantified by a BCA protein assay reagent kit at 562 nm [43]. Independent tests were performed in triplicate samples.

Contact-active antibacterial activity: 50 μL WPUn emulsions were casted onto cover glasses (1.5 × 1.5 cm^2^) and dried completely. Then, all these samples were immersed in water at 37 °C for 6 h and dried to exclude the effects of small molecular antibacterial components. After 2 h UV-sterilization, these samples were sprayed with an aqueous suspension of *S. aureus* (1 × 10^6^ cells·mL^−1^) and air-dried for 10 min, then incubated in nutrient broth agar (0.8% agar) media at 37 °C for 16–24 h. Bacterial colonies grew from the individual cells and stained red with 3 mL triphenyltetrazolium chloride (5 mg·mL^−1^).

Evaluation of antibiofilm activities of WPUn films: To eliminate the influences of unreacted PEG and GQAS moieties onto WPUn films of their anti-fouling activities, all the films were immersed in water (110 rpm, 37 °C) for 6 h before the test. The anti-fouling activities of these films were performed against both *E. coli* and *S. aureus* according to shaking flask methods [44]. Bacteria strains were grown overnight at 37 °C in culture media (LB + casamino acids for *E. coli* [45] and TSB+ 2% glucose for *S. aureus* [46,47]), and then diluted to 10^7^ cells·mL^−1^. Each WPU film (1.0 × 1.0 cm^2^, 0.5 mm thickness) was placed in a 24-well plate and sterilized under UV overnight. A total of 2 mL bacteria liquid was added into each well with one sample and cultured at 37 °C, 110 rpm for 2 days. After that, tested films were carried out, rinsed with sterile deionized water, and divided into two groups. One group was placed into a centrifuge tube with 2 mL of sterile water. The bacteria were detached from WPUn films in an ultrasonic cleaner for 5 min and counted by the method of agar dish. The other group was fixed with glutaraldehyde and dried, then the morphology of these WPUn surfaces and the attached bacteria were examined using a scanning electron microscope. Tests using LB culture media without casamino acids, or TSB without glucose, were also performed as a control. All the tests were replicated at least three times.

### 2.5. Cytotoxicity Assay

Cytotoxicity of WPUn film extracts and degradation products were performed by Methylthiazoletetrazolium (MTT) assay with a similar process to our previous work [32]. Latex rubber film (Microflex Corp., Sparks, NV) was selected as positive control. Degradation media (NaOH + HCl) were also tested as a blank control. The cell viability was obtained by the following equation.
$$Cell\;viability\;\left(\%\right)=\frac{OD_{sample}}{OD_{negative}}\times 100\;$$

### 2.6. Statistical Analysis

Data are expressed as means (standard deviations). A statistical analysis was made by the Statistical Package for the Social Sciences (SPSS 22.0) software (IBM, Armonk, NY, USA).

### 2.7. Characterization

Proton nuclear magnetic resonance (^1^H NMR, 400 MHz) spectra were obtained on a Bruker AV II-400 MHz spectrometer in DMSO (Bruker, Billerica, MA, USA).

Transmission Fourier transform infrared (FTIR) spectra were tested at 25 °C on a Nicolet-6700 spectrophotometer (Thermo Electron Corporation, Waltham, MA, USA) between 4000 and 600 cm^−1^ (resolution of 4 cm^−1^).

Gel permeation chromatography (GPC) was performed by Waters-1515 (Waters, Milford, CT, USA) using N,N-dimethylformamide/LiBr as eluent, and polymethyl methacrylate as reference. Test concentration of WPUn samples was 2–3 mg·mL^−1^, and the flow rate was 1 mL·min^−1^ at 40 °C.

Differential scanning calorimetry (DSC) was performed on a Netzsch STA 449C Jupiter (Netzsch, Selb, Germany). The heating rate was 10 °C·min^−1^ in the range of −120 to 100 °C under a steady flow of nitrogen.

Zeta potential of WPUn emulsions (diluted with distilled water to about 0.02 wt% before the test) were tested using a Zetasizer Nano ZS dynamic light-scattering (DLS) instrument (Malvern, Worcestershire, UK) at room temperature at an angle of 90°.

Water contact angles (WCA) of WPUn surface were measured by a Drop Shape Analysis System DSA 100 (Kruss, Hamburg, Germany). Measuring parameter: 3 μL of distilled water at 25 °C. The results were the mean values of three replicates.

## 3. Results and Discussion

### 3.1. Preparation and Characterization of WPUn Emulsions

Self-polishing polyurethanes with different content of GQAS have been prepared. WPU0 without GQAS was prepared as a comparison. The molecular weights (Mn) of the resulting WPUn are increased after introducing EG12 (62419 to 93699) (Table 1). However, the more EG12 introduced, the lower Mn of the resulting WPUn (96399 to 39613). This can be explained by the following two reasons. (1) In the synthesis process of waterborne polyurethane, the two-step chain extension will react completely. (2) Though lysine and EG12 both have diamine reactive groups, lysine with a simple structure has higher reaction efficiency than EG12 with pendent GQAS groups. Since dilution may affect the emulsion stability [48], zeta potential of these WPUn emulsions (Table 1) obtaining from DLS is only for comparing the behavior of this system. Zeta potential of these WPUn emulsions is shown in Table 1. Zeta potential of WPU0 without EG12 is –4.92 mV, which is due to the carboxylate groups of lysine through neutralization (Table 1). With the introduction of EG12, Zeta potential of WPU8.5 emulsion changes from negative to positive, despite there being more anionic groups than cationic groups in their structure. In addition, the more the chain extender EG12 in the corresponding WPUn, the higher the zeta potential of WPUn emulsions. This indicates that GQAS with high interface energy tends to be aggregated on the particle surface, that is, the interface between the particle and water. This aggregation behavior is beneficial for the antibacterial properties of these WPUn emulsions, which will be discussed in the following part.

In the ^1^H NMR spectra (Appendix A), peaks at 3.04–3.31 ppm are attributed to the methylene and methyl groups connected with N^+^ that can only be found in WPUn samples containing EG12, and their intensity increased with the increase in EG12 content. Peaks of 1–8 are ascribed to IPDI, while peak at 3.53 ppm belongs to the methylene groups of PEG. Peaks at 1.26, 1.55, 2.27, and 3.99 ppm are ascribed to methylene protons of PCL. A peak at 3.81 ppm belongs to the terminal methylene group in PCL and PEG. Additionally, the FTIR spectra of WPUn were shown in Figure 2A. Absorption peaks at 3170–3550 cm^−1^ are ascribed to N–H vibration in urea and urethane structures. Absorption bonds approximately at 3400–3500 cm^−1^ are attributed to non-hydrogen bonded N–H group vibrations, which are barely observed, indicating that most N–H groups in WPUn are involved in hydrogen bonds (hydrogen bonded N–H appears at around 3330 cm^−1^) [49]. Absorption bands at 2944 and 2865 cm^−1^ correspond to C–H stretching vibrations. In the carbonyl region, adsorption peak around 1655 cm^−1^ belongs to the carbonyl in urea groups from EG12 and lysine, while peaks at around 1725 cm^−1^ are assigned to carbonyl in urethane groups and PCL. These results indicate that these WPUn samples have been successfully prepared.

Thermal behavior of these obtained WPUn samples was conducted using DSC (Table 1 and Figure 2B). The glass transition temperatures (T_g_) of soft segments of all WPUn ranged from −55.12 to −55.52 °C, and have only tiny changes, which means that the introduction and content of GQAS hardly change the aggregation state of WPUn. Additionally, T_g_ of these WPUn are slightly higher than that of pure PCL (−58 °C) [50], meaning high degree phase separation of hard-soft segment in these WPUn matrices. This phenomenon can be explained that the strong polarity difference between soft and hard segments leading to the microphase separation in these WPUn. The low T_g_ of WPUn samples made them good candidates to be served in the low-temperature environment. 

### 3.2. Degradation Properties of WPUn

Under the requirements of environmental protection, coatings decomposed and degraded to low-toxic or non-toxic components will be of great significance. What is more, the degradation pattern with layered degradation is considered as extremely promising and efficient self-polishing method, be it that the attached living organisms could be released along with the decomposition and degradation. Thus, the hydrolytic and enzymatic degradation behavior of these obtained WPUn films were performed by measuring their weight losses at setting time (Figure 3). The weight loss ratios of these obtained WPUn increased sharply in the first 6 h, and then slowed down for both hydrolytic and enzymatic degradation. This is because that the good solubility and low molecular weight of hydrophilic oligomer in WPUn can leach out after immersing in degradation media at the initial stage. In addition, the degradation rates of these WPUn in PBS + lipase PS media were higher than those in only PBS media, because lipase PS can promote the decomposition reaction of PCL-based polymers [32]. In addition, the degradation rates of WPUn films increased along with the increase in GQAS. The probable reason is that the more contents of GOAS, the more hydrophilic of these WPUn surfaces (Figure 4a), and the more likely to be dissolved and degraded by lipase PS. 

SEM was used to observe the surface morphology changes of these WPUn samples after degradation to study their degradation mechanism. Figure 4 shows the SEM images of WPU film surfaces without degradation, hydrolytic degradation in PBS, and enzymatic degradation in PBS + lipase PS media. After 28 days of degradation in PBS, surface of WPUn shows lots of holes and even cracks (Figure 4(a1,b1,c1,d1)), which might because the water-soluble oligomers or monomers formed from amorphous areas of WPUn after hydrolytic degradation and then leached out into the surrounding media. Different from hydrolytic degradation, enzymatic degradation shows a superficial decomposition without any pores or cracks under lipase PS conditions (Figure 4(a2,b2,c2)). However, with the increase in GQAS content, WPU30 also formed cracks surface morphology (Figure 4(d2)), which might be because that WPU30 with low molecular weight and high hydrophilic components are more liable to be dissolved even in the lipase PS environment. Carefully observing the surface morphology of WPU30, differences still can be found in that there is no porous structure on the surface except cracks under enzymatic degradation compared with hydrolysis degradation, which is mainly caused by the different degradation mechanisms of hydrolysis and enzymatic degradation. Hydrolysis of PCL-based polyurethane might mainly start from the amorphous PCL ester bond in the soft segment, and then gradually deepen to form holes. When enzymes are present, they can adsorb onto the WPU surface before the hydrolysis, promote rapid degradation, and eventually appear as a surface erosion mechanism. This surface erosion mechanism can give WPU films efficient anti-fouling properties by releasing the attached bacteria along with the degradation. In addition, these results combined with our previous work [32] demonstrate the surface erosion mechanism of WPUn films in the presence of enzymes, which made them good anti-fouling and self-polishing abilities. However, when the content of GQAS reaches to 30 wt% content, or the hydrophobic alkyl chain length is too long, cracks and holes will be appeared in the process of both hydrolytic and enzymatic degradation, which go against to the anti-fouling of WPUn films.

### 3.3. Anti-Fouling Efficacy

#### 3.3.1. MIC Studies of WPUn Emulsions

To test the antibacterial properties of the obtained WPUn emulsions, MIC assay was performed against suspensions *E. coli* (gram-negative) and *S. aureus* (gram-positive), using EG12 as control. The MICs of waterborne WPUn containing GQAS ranged 11.25–98.85 µg·mL^−1^ for *S. aureus*, and 44.90–197.65 µg·mL^−1^ for *E. coli*, representing good antibacterial activity (Table 2). In addition, we found that the EG12 concentration (corresponding EG12 concentration: 3.2–8.4 µg·mL^−1^ for *S. aureus*, and 10.6–16.8 µg·mL^−1^ for *E. coli*) in WPUn emulsions at the concentration of MIC, was smaller than the MIC of pure EG12 (4 µg·mL^−1^ for *S. aureus*, 16 µg·mL^−1^ for *E. coli*). A possible reason was that cationic GQAS aggregated on the surface of the WPUn emulsion particles, as demonstrated by DLS (Table 1). These samples and EG12 show more efficiency against *S. aureus* than to *E. coli*, because of the very sophisticated outer cell wall of *E. coli* that effectively keeps out antibacterial agents. The positive zeta potential (1.07–5.40) of WPUn emulsions enhances the interaction between the surface of nanoparticles and bacterial membrane, resulting in good antibacterial activity [31] (Table 1).

#### 3.3.2. Hydrophilic Analysis of WPUn Films

Super-hydrophobic or hydrophilic surface can prevent protein adsorption, and then prevent fouling organisms from contact and adhesion. Therefore, the hydrophilicity of these WPUn coatings was determined by contact angle (WCA) measurements. A decrease in WCA was measured for all samples after contact with water (Figure 5A). There are two reasons that could account for the decrease in WCA. One is because that after contact with H_2_O, the new hydrogen bonding interacts between PEG, or carboxyl and H_2_O would promote their migration onto the surfaces [51]. Furthermore, the contact angles of WPUn containing GQAS are much lower than those of WPU0, which are attributed to the cationic nature of QAS polymers on their surfaces [50]. Thus, a reduction in WCA is expected.

#### 3.3.3. Protein Adsorption Assay

A surface with persistent protein-resistant property can polish away adhered fouling, including inorganics or organics, resulting in good anti-fouling performance in complex environment [52]. Nonspecific protein adsorption is a key factor for anti-fouling activity of material because the formation of biofilm begins with the adhesion of proteins and glycoproteins. Therefore, the protein adsorption assay was performed in this study, and the results were illustrated in Figure 5B. The amount of protein adsorption on control film TPEU surface (without PEG and GQAS) is about 1.3 mg/cm^2^, while only 0.2 mg/cm^2^ on WPU0 surfaces. This might be due to the repulsion of –COOH and the rearrangement of PEG after contact with water, which can reduce the protein adsorption. However, the amount of protein adsorption on the surfaces of WPUn increases with the increase in GQAS contents. This is because the increase in cationic GQAS could convert WPUn surface from negatively charged to positively charged, and enhance electrostatic interaction between the surface and BSA. In addition, the amount protein adsorption for the WPUn films was lower under the shaking state than the stationary state. Therefore, these WPU films would have better performance in the flowing liquid environment.

#### 3.3.4. Contact-Active Antibacterial Activity of WPUn Films

Contact-active antibacterial activity of WPUn films was performed by a glass slide spreading method. There were no bacteria colonies observed in films containing GQAS, meaning good contact-active antibacterial activity (Figure 5C). In addition, no inhibition zone was observed around WPUn films, indicating that almost no GQAS leaching out from the coatings. Meanwhile, lots of bacteria colonies were observed on WPU0 film surface, indicating that a little GQAS being introduced can give WPU films good contact-active antibacterial activity.

#### 3.3.5. Anti-Biofilm Assay

Biofilms are known to be one of the most hard-to-treat bacterial pollutions, through protecting bacteria from antibacterial agents, and dispersing planktonic bacteria to distant sites. Therefore, in this work, cultures with biofilm formation factors that can promote the formation of bacterial biofilms (Luria Bbertani broth supplemented with 0.5% casamino acids for *E. coli*, TSB + 2% glucose for *S. aureus*) were designed as a model to study the anti-biofilm properties of these WPU films. Common cultures without biofilm formation factors were used as a comparison, and the results were recorded in Figure 5D and Figure 6. After introducing biofilm formation factor, adhering live bacteria on WPU0 surface increased about 5 times (*S. aureus*) and 13 (*E. coli*) times than that in common culture, meaning casamino acids and glucose indeed promote the biofilm formation. Only the content of GQAS in polyurethane reached 15.8 wt%, these obtained WPUn films can prevent *E. coli* biofilm formation (Figure 5D), while only 8.5 wt% can prevent *S. aureus* biofilm formation. There are two reasons that can explain this phenomenon. One is possibly because of the antibacterial mechanism of GQAS, that the cationic headgroup of GQAS attracts anionic bacterial membrane after contacting, then the long hydrophobic alkyl chains of GQAS will pierce the membranes of attached bacteria, causing cytoplasm leakage, lysis, and death. Gram-negative bacteria have an extra extracellular membrane than Gram-positive bacteria [53,54], which can protect them against the harmful effects of GQAS. Therefore, the antibacterial effect of polymer-based GQAS on Gram-positive bacteria is generally higher than that on Gram-negative bacteria. The other reason might be because of the surface erosion mechanism of WPUn films, which made them good self-polishing abilities and persistent antibacterial effect.

### 3.4. Cytotoxicity of WPUn Films

Security is critical for materials’ applications. Thus, the cytotoxicity of these WPUn films were evaluated by testing the cytotoxicity of their extracts and degradation liquor. The effect of WPUn extracts on the proliferation of L929 mouse fibroblasts is shown in Figure 7. The cytotoxicity against fibroblasts increased with the increase in GQAS concentration in polyurethanes. However, more than 80% cell viability is retained after five-times dilution after 72 h incubation of all WPUn samples (Figure 7B). Compared with the positive sample (latex rubber) with less than 20% cell viability even after 100 times of dilution, all the WPUn samples show much lower cytotoxicity.

Cell viability of WPUn degradation products with different concentrations after 24 h and 72 h incubation was studied and illustrated in Figure 8. The results show that there is no apparent inhibitory effect for the L929 fibroblasts when the concentration of WPUn degradation solutions diluting to 6.3 mg·mL^−1^ after 24 h incubation (Figure 8). In addition, more than 70% cell viability is still retained when diluting WPUn degradation solution to 2.5 mg·mL^−1^ after 72 h incubation. There are two reasons that can explain the low cell viability of WPUn degradation solutions at a high concentration (>6.3 mg·mL^−1^). One is the inherent toxicity of polycations [55]. The other reason is the large amounts of sodium chloride in the degradation media (NaOH + HCl), in that the blank control without dilution also shows low cell viability (Figure 8). Therefore, in consideration of actual usage, these WPUn films will potentially meet safety requirements of food storage and packaging, protect coatings, water purification systems, textiles, biomedical, and marine engineering applications.

## 4. Conclusions

In conclusion, a series of self-polishing waterborne polyurethanes containing different amounts of GQAS with excellent antibacterial activity, anti-biofilm activity, and good biocompatibility were successfully synthesized. These WPUn films can lead to hydrolytic and enzymatic degradation. Both degradation rates increase with EG12 content. It is worth noting that WPUn films can lead to layered degradation in the presence of enzymes, which made them good anti-fouling and self-polishing capabilities. However, when the content of GQAS reached to certain content (30 wt%), or the hydrophobic alkyl chain lengths were too long, cracks and holes appeared in the process of both hydrolytic and enzymatic degradation, which adversely affect the anti-fouling properties of WPUn films. These WPUn films also exhibit excellent anti-biofilm activity when the content of GQAS reached to 8.5 wt% against *S. aureus*, and 15.8 wt% against *E. coli*. The cytotoxicity of these obtained polyurethanes was also systematically studied; they could be degraded to non-toxic or low toxic compositions. Thus, these biodegradable self-polishing and anti-fouling WPUn can be potentially applied as coatings for surgical equipment, health care, and marine engineering to prevent microbial contamination.

## Figures and Tables

**Figure 1 polymers-15-00317-f001:**
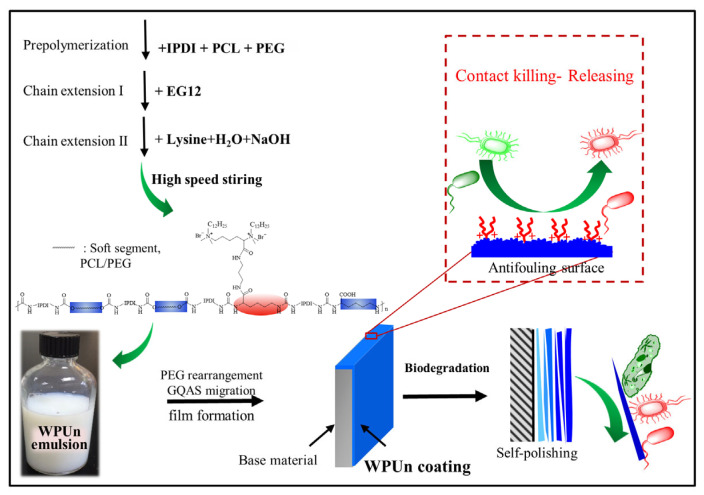
Schematic structure of WPUn coatings with anti-fouling surface.

**Figure 2 polymers-15-00317-f002:**
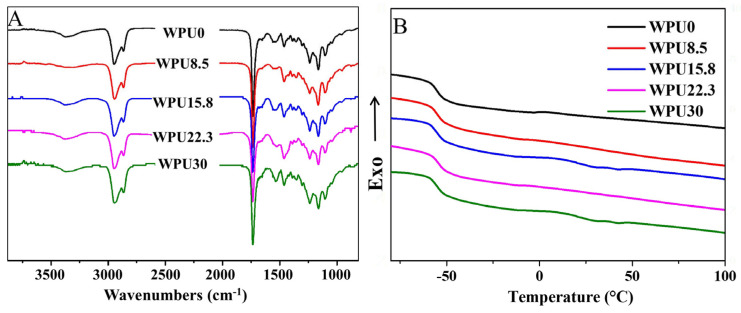
(**A**) The transmission FTIR spectra of WPUn samples. (**B**) DSC thermograms of WPUn samples (second heating).

**Figure 3 polymers-15-00317-f003:**
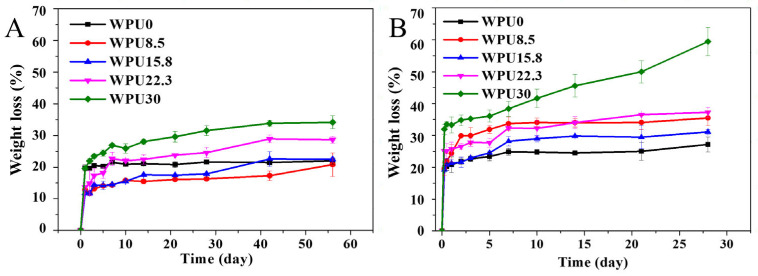
Weight loss profile of WPUn films under hydrolytic and enzymatic degradation. (**A**) Hydrolytic degradation in PBS; (**B**) Enzymatic degradation in PBS + lipase PS.

**Figure 4 polymers-15-00317-f004:**
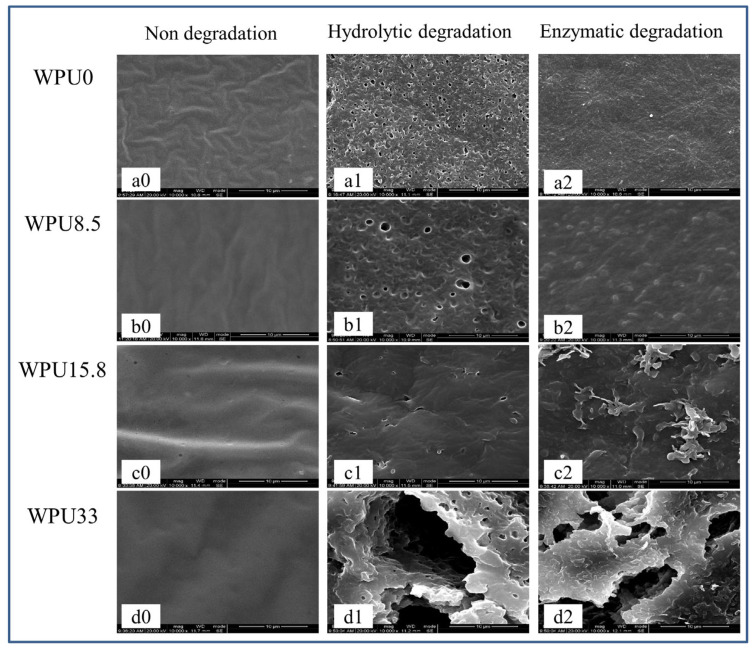
Surface morphology of WPUn. (**a0**,**b0**,**c0**,**d0**): WPUn non-degraded; (**a1**,**b1**,**c1**,**d1**): WPUn degraded in PBS; (**a2**,**b2**,**c2**,**d2**): WPUn degraded in PBS + lipase PS. Scale bar = 10 μm.

**Figure 5 polymers-15-00317-f005:**
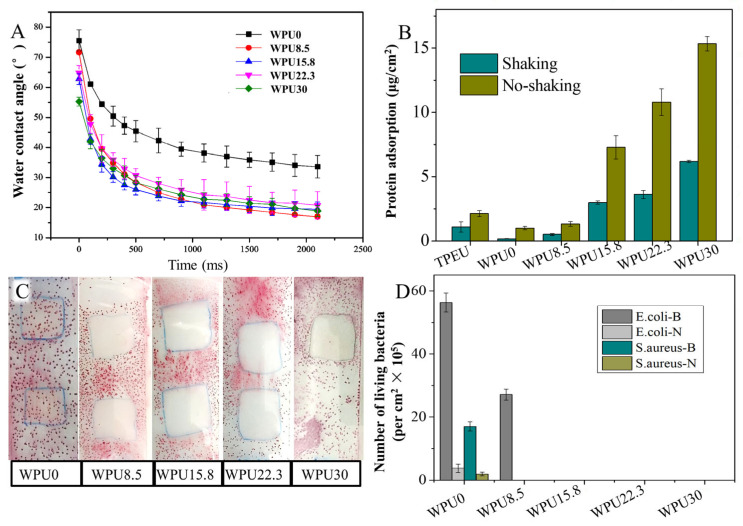
Anti-fouling activity of WPUn films. (**A**) Time-related water contact angle of WPUn films; (**B**) Nonspecific protein adsorption of WPUn films; (**C**) Contact-active antibacterial activity of WPUn films; (**D**) Number of living bacteria attached on WPUn under different bacterial culture (*E. coli*-LB+: LB + 0.5%vcasamino acids; *E. coli*-LB: LB; *S. aureus*-TSB+: (TSB + 2% glucose; *S. aureus*-TSB: TSB).

**Figure 6 polymers-15-00317-f006:**
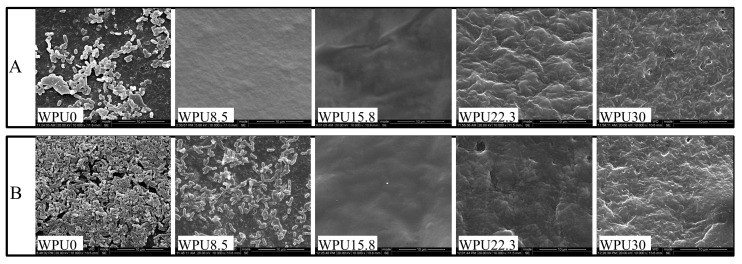
Morphology of *E. coli* attached to WPUn films (10,000×) after 2 days incubation in *E. coli* culture (10^7^ CFU·mL^−1^). (**A**) LB media; (**B**) LB + 0.5% casamino acids media. Scale bar = 10 μm.

**Figure 7 polymers-15-00317-f007:**
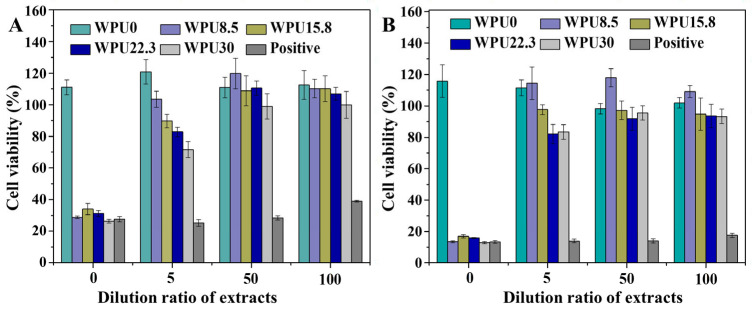
Cell viability of WPUn extracts with serial dilutions after 24 h (**A**) and 72 h (**B**) incubation. Error bars represent means ± standard deviation for n = 3. Statistical significance: *p* < 0.05.

**Figure 8 polymers-15-00317-f008:**
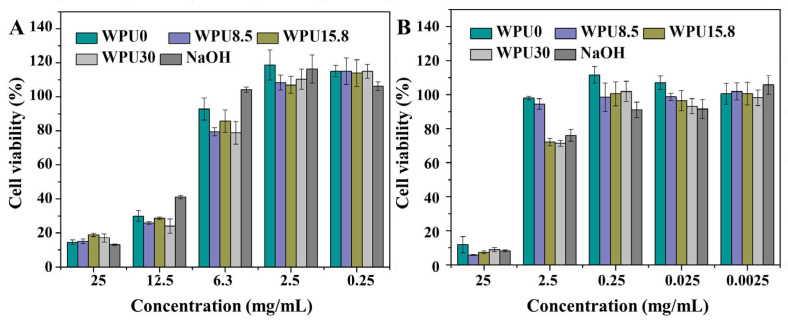
Cell viability of WPUn degradation products with different concentrations after 24 h (**A**) and 72 h (**B**) incubation. Error bars represent means ± standard deviation for n = 3. Statistical significance: *p* < 0.05.

**Table 1 polymers-15-00317-t001:** The theoretical composition, molecular weight, Zeta potential, and glass transition temperature (Tgs) of WPUn samples.

Samples	Molar Ratio of IPDI/PEG/PCL/EG12/Lysine	Dosage of EG12 (wt%)	Mn (g/mol)	Mw/Mn	Zeta Potential (mV)	Tg (°C)
WPU0	2.05:0.33:0.67:0:1	0	62,419	1.41	−4.92 ± 0.38	−55.27
WPU8.5	2.05:0.33:0.67:0.25:0.75	8.5	93,688	2.09	1.07 ± 0.49	−55.52
WPU15.8	2.05:0.33:0.67:0.5:0.5	15.8	55,265	1.53	2.89 ± 0.79	−55.34
WPU22.3	2.05:0.33:0.67:0.75:0.25	22.3	39,613	1.40	3.68 ± 0.36	−55.39
WPU30	2.05:0.33:0.67:1:0	30.0	39,251	1.53	5.40 ± 0.88	−55.12

**Table 2 polymers-15-00317-t002:** MICs of WPUn emulsions, EG12, and DTAB (µg·mL^−1^).

Samples	*E. coli*	*S. aureus*
WPU0	>8000 (0 *)	>8000 (0 *)
WPU8.5	197.65 (16.8 *)	98.85 (8.4 *)
WPU15.8	91.40 (14.4 *)	45.7 (7.2 *)
WPU22.3	47.65 (10.6 *)	23.85 (5.3 *)
WPU30	44.90 (12.6 *)	11.25 (3.2 *)
EG12	16.00	4.00
DTAB	64.00	16.00

* The converted concentration of EG12: (µg·mL^−1^).

## Data Availability

The data presented in this study are available on request from the corresponding author.

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
