# Peer review of "Anti-Fouling and Anti-Biofilm Performance of Self-Polishing Waterborne Polyurethane with Gemini Quaternary Ammonium Salts"

_polymers, 2023, doi:10.3390/polym15020317_

Round 1
Reviewer 1 Report
The manuscript reports the Preparation of eco-friendly self-polishing coatings along with Anti-fouling Behavior. The manuscript is well written and organized; however, revision is needed according to the following comments.
1) The novelty of the work needs to be highlighted clearly in the introduction as compared to other articles published recently in the year 2020-2022.
2) Sub-section 2.2: Authors should re-write this section to elaborate the procedure of coating preparation in detail with scientific explanation of ratio that was selected by authors. This will be better for proper understanding of readers.
3) Scale bar should be reported the SEM images along with date of analysis Fig. 3 and 5.
4) The stability of developed coatings should be explained under harsh environmental condition such as various ranges of pHs and temperatures.
5) The mechanism of antifouling efficacy and anti-bacterial behaviour should be provided with more clear scientific explanation with the use some more characterization analysis. This will be useful to reproduce the such coatings for further research.
6) Quality of figures are very poor, should be improved in revised manuscript.
7) English of the manuscript should be revised carefully.
Therefore, my recommendation is resubmission of the manuscript with a major revision in the suggested points, before being considered for publication in this Journal.
Author Response
Response to the comments of Referee #1:
1) The novelty of the work needs to be highlighted clearly in the introduction as compared to other articles published recently in the year 2020-2022.
Response: The manuscript has been revised accordingly. Articles published recently in the year 2020-2022 has been added and discussed in the introduction as follows.
“Vamvakaki, et al., prepared biocompatible antimicrobial hybrid coatings combining enhanced biocidal activity and stimuli-renewable antimicrobial action based on a natural polymer, chitosan, modified to bear QAS groups and modified TiO2 nanoparticles [28].” (Line 85-88).
“Self-polishing antifouling coating could also be an ideal choice in antifouling ap-plications[35]. Zhou et al., developed a series of interpenetrating polymer networks (IPNs) equipped with zwitterionic nanoparticles and lubricant for quasistatic and dy-namic antifouling[36]. Zhou et al., prepared a self-polishing antifouling coating via a borneol/ boron acrylate polymers, which can remove the adhered biofouling by hydrol-ysis to renew the surface [37].” (Line 101-106).
[28] T. Manouras, E. Koufakis, E. Vasilaki, I. Peraki, M. Vamvakaki, Antimicrobial Hybrid Coatings Combining Enhanced Biocidal Activity under Visible-Light Irradiation with Stimuli-Renewable Properties, ACS Applied Materials & Interfaces 13(5) (2021) 17183-17195.
[35] H. Zhang, J. Cao, L. Sun, F. Kong, J. Tang, X. Zhao, Y. Tang, Y. Zuo, Comparative Study on the Degradation of Two Self-Polishing Antifouling Coating Systems with Copper-Based Antifouling Agents, Coatings 12(8) (2022) 1156.
[36] J. Zhang, X. Wang, C. Zhang, Self-lubricating interpenetrating polymer networks with functionalized nanoparticles enhancement for quasi-static and dynamic antifouling, Chemical engineering journal (2) (2022) 429.
[37] F. Song, J. Wang, L. Zhang, R. Chen, Q. Liu, J. Liu, J. Yu, P. Liu, J. Duan, Synergistically Improved Antifouling Efficiency of a Bioinspired Self-renewing Interface via a Borneol/ Boron Acrylate Polymer, Journal of Colloid and Interface Science 612 (2022) 459-466.
2) Sub-section 2.2: Authors should re-write this section to elaborate the procedure of coating preparation in detail with scientific explanation of ratio that was selected by authors. This will be better for proper understanding of readers.
Response: The manuscript has been revised accordingly. The sentence
“The feed ratios were shown in Table 1.”
has been revised to
“The feed ratios were shown in Table 1. The content of GQAS in WPUn structure was regulated by changing the ratio of two chain extenders” (Line 142-144).
The sentence
“WPUn films were prepared with a similar process to our previous work [32] before testing, including antibacterial, anti-biofilm activity and cytotoxicity.”
has been revised to
“WPUn coatings were prepared with a similar process to our previous work [33] , by casting these WPUn emulsions siliconized culture dishes and dried at room temperature for 2 d, 60 ºC for 2 d, and then 60 ºC under vacuum for 2 d. The films were cut in sheets (1×1 cm, 0.5 mm thickness), and used directly for tests unless mentioned otherwise. To eliminate the influences of unreacted PEG and GQAS moieties onto WPUn films of their anti-fouling activities, all the films were immersed in water (110 rpm, 37 ºC) for 6h, then put into an oven at 60 ºC for 2 d, before testing, including antibacterial, anti-biofilm activity, protein adsorption assay, and cell cytotoxicity.” (Line 152-159).
3) Scale bar should be reported the SEM images along with date of analysis Fig. 3 and 5.
Response: Scale bar has been added in caption of Figure 3 and 5 accordingly.
The caption of Figure 3
“Surface morphology of WPUn. a0-d0: WPUn non-degraded; a1-d1: WPUn degraded in PBS; a2-d2: WPUn degraded in PBS + lipase PS”
has been revised to read
“Surface morphology of WPUn. a0-d0: WPUn non-degraded; a1-d1: WPUn degraded in PBS; a2-d2: WPUn degraded in PBS + lipase PS. Scale bar = 10 μm.” (Line 305-306).
The caption of Figure 5
“Morphology of E. coli attached to WPUn films (10000 ×) after 2 days incubation in E. coli culture (107CFU·mL-1). (a) LB media; (b) LB+0.5% casamino acids media”
has been revised to read
“Morphology of E. coli attached to WPUn films (10000 ×) after 2 days incubation in E. coli culture (107CFU·mL-1). (a) LB media; (b) LB+0.5% casamino acids media. Scale bar = 10 μm.” (Line 415-416).
4) The stability of developed coatings should be explained under harsh environmental condition such as various ranges of pHs and temperatures.
Response: We thank the reviewer for the critical comments and helpful suggestions. Experiments about the stability of these WPUn coatings under harsh environmental condition such as various ranges of pHs and temperatures has not performed because of the COVID-19 pandemic in Guangzhou recently. However, the degradation property of waterborne polyurethane with a similar structure was studied under different pHs, that the polyurethane displays a low degradation in acidic PBS solutions. The degradation of these polyurethanes containing PCL mainly occurs by the breakdown of ester linkages in PCL segments.
5) The mechanism of antifouling efficacy and anti-bacterial behaviour should be provided with more clear scientific explanation with the use some more characterization analysis. This will be useful to reproduce the such coatings for further research.
Response: We thank the reviewer for the critical comments and helpful suggestions. The sentence
“This possibly because of the antibacterial mechanism of GQAS, that the cationic head-group of GQAS attracts anionic bacterial membrane after contacting, then the long hydrophobic alkyl chains of GQAS will pierce the membranes of attached bacteria, causing cytoplasm leakage, lysis, and death. Gram-negative bacteria have an extra extracellular membrane than Gram-positive bacteria [53, 54], which can protect them against the harmful effects of GQAS. Therefore, the antibacterial effect of polymers-based GQAS on Gram-positive bacteria is generally higher than that on Gram-negative bacteria.”
has been revised to
“There are two reasons that can explain this phenomenon. One is possibly because of the antibacterial mechanism of GQAS, that the cationic headgroup of GQAS attracts anionic bacterial membrane after contacting, then the long hydrophobic alkyl chains of GQAS will pierce the membranes of attached bacteria, causing cytoplasm leakage, lysis, and death. Gram-negative bacteria have an extra extracellular membrane than Gram-positive bacteria [53, 54], which can protect them against the harmful effects of GQAS. Therefore, the antibacterial effect of polymers-based GQAS on Gram-positive bacteria is generally higher than that on Gram-negative bacteria. The other reason might be because of the surface erosion mechanism of WPUn films, which made them good self-polishing abili-ties and persistent antibacterial effect.” (Line 404-413).
6) Quality of figures are very poor, should be improved in revised manuscript.
Response: Quality of all figures have been improved in revised manuscript accordingly.
7) English of the manuscript should be revised carefully.
Response: The manuscript has been revised accordingly. The whole manuscript has been checked thoroughly by a native-English speaker.

Reviewer 2 Report
The Authors presented the results of the system based on modified polyurethane which could be applied as an active coating. The idea of the presented results is interesting, however, a lot of improvements in the manuscript content should be done.
First of all, I m not convinced that the degradation was fully proven - did you check the post-products? Then a lot of results are missing, i.e. DSC, TGA, and DLS curves. There is no explanation of how the samples for DLS were prepared. Even more, it is not clear in which form, the samples were studied - emulsion, film, coat, etc. It should be highlighted. For that reason, major revision is needed.
Detailed comments are listed below:
Abstract
- the first sentence is difficult to catch the meaning
- lines 15 - 19 this description is too general and more suitable for introduction or results and discussion section
- in the abstract please pay more attention to the most interesting results and the methods which were applied
- please clearly state what was achieved - emulsion/film/blend, etc.
Introduction
- figure 1 should be presented in the materials and methods section
- please clarify why the system is eco-friendly (usually polyurethanes are not classified as "eco"
- did you examine biodegradability
Materials and methods
- brief description of EG12 preparation is necessary
-In section 2.2 please avoid statements like eco-friendly etc. Just give state of art such as emulsion preparation
- the description of the degradation is difficult to understand
- please clearly state how the experiments were done - in emulsions or did you prepare films from the emulsions (how?)
Results and discussion
- line 222 and others please avoid eco-friendly where it is not fully proven
- how the emulsions were studied - did you dilute them? What about the particle size distribution?
- did you verify what was the products of the degradation?
- figure 3 please magnify the scale bar
- section 3.3.4. how did you verify no bacteria, even from the air?
- why there are no curves from DSC and TGA in the manuscript?
Author Response
Abstract
1) the first sentence is difficult to catch the meaning
Response: The manuscript has been revised accordingly. The sentence
“Biofilms are known to complicate most of the hard-to-treat bacterial pollution in medical, public health and marine areas.”
has been revised to
“Biofilms are known to be difficult to eradicate and control, complicating human infections and marine biofouling.” (Line14-15).
2) lines 15 - 19 this description is too general and more suitable for introduction or results and discussion section
Response: The manuscript has been revised accordingly. The sentence
“In this study, a series of novel eco-friendly self-polishing waterborne polyurethane emulsions with different content lysine-derivative gemini quaternary ammonium salts (GQAS) are synthesized. The resulting waterborne polyurethanes can both hydrolytically and enzymatically degrade, especially under enzyme conditions.”
has been revised to
“In this study, self-polishing and anti-fouling waterborne polyurethane coatings synthesized from gemini quaternary ammonium salts (GQAS), polyethylene glycol (PEG), and polycaprolactone diol (PCL) demonstrate excellent antibiofilm efficacy.” (Line15-17).
3) in the abstract please pay more attention to the most interesting results and the methods which were applied
Response: The manuscript has been revised accordingly. The sentence
“What’s more, the surface erosion enzymatic degradation mechanism made them good self-polishing ability. Both hydrolytic and enzymatic degradation rates increase with the increase of GQAS contents. Their anti-fouling and anti-biofilm performance was confirmed by a cul-ture-based method in broth media with biofilm formation factor against Gram-positive (S. aureus) and Gram-negative bacterial strains (E. coli) for 2 days. The results indicate that polyurethane coatings have excellent anti-biofilm activity when the content of GQAS reached to 8.5 wt% against S. aureus, and 15.8 wt% against E. coli. The cytotoxicity of these obtained polyurethanes’ extracts and degradation liquids was also systematically studied. These results indicate that these self-polishing WPUn coatings with GQAS possess good anti-biofilm abilities, and biocompatibil-ity, and potentially applied in bio-medical and marine environmental applications.”
has been revised to
“Their anti-fouling and anti-biofilm performance was confirmed by a culture-based method in broth media, with the biofilm formation factor against Gram-positive (S. aureus) and Gram-negative bacterial strains (E. coli) for 2 days. The results indicate that polyurethane coatings have excellent anti-biofilm activity when the content of GQAS reached 8.5 wt% against S. aureus, and 15.8 wt% against E. coli. The resulting waterborne polyurethane coatings demonstrate both hydrolytic and enzymatic degradation, and the surface erosion enzymatic degradation mechanism enable them with good self-polishing capability. The extracts cyto-toxicity of these polyurethane coatings and degradation liquids was also systematically studied, they could be degraded to non-toxic or low toxic compositions. This study shows the possibility to achieve potent self-polishing and anti-biofilm efficacy by integrating antibacterial GQAS, PEG and PCL into waterborne polyurethane coatings.” (Line17-27).
- please clearly state what was achieved - emulsion/film/blend, etc.
Response: We thank the reviewer for the critical comments and helpful suggestions. Waterborne polyurethane coating was achieved and clearly state in the abstract. (Line15).
Introduction
- figure 1 should be presented in the materials and methods section
Response: The manuscript has been revised accordingly. Figure 1 has been moved to the materials and methods section. (Line 160-161)
2) please clarify why the system is eco-friendly (usually polyurethanes are not classified as "eco" did you examine biodegradability
Response: We thank the reviewer for the critical comments and helpful suggestions. The word “eco-friendly” has been removed in the manuscript.
Materials and methods
- brief description of EG12 preparation is necessary
Response: We thank the reviewer for the critical comments and helpful suggestions. The synthetic route of EG12 has been added in the Supplementary information as follows (Manuscript, line 132-133; Supplementary Material, Page 1).
Scheme S1. The synthesis route of gemini quaternary ammonium (GQAS) chain extender (EG12).
- In section 2.2 please avoid statements like eco-friendly etc. Just give state of art such as emulsion preparation
Response: The manuscript has been revised accordingly. The sentence
“2.2 Preparation of eco-friendly self-polishing WPUn emulsions and coatings
A series of eco-friendly self-polishing WPUn emulsions based on IPDI, PCL, PEG, lysine and EG12 were synthesized by two-step polymerization. The feed ratios were shown in Table 1. PEG and PCL were dehydrated under vacuum at 80-90 ºC for 2 h firstly. Then IPDI and 0.1% stannous octoate were added under N2 environment. After contin-uous reaction for other 2 h at 70-75 ℃, the reaction prepolymer was cooled down. Then different content of chain extender EG12 in acetone was added in the mixture for 0.5 h. Afterward, the polyurethane prepolymer was poured into L-lysine aqueous solution under stirring (600 rpm) and ultrasound (200W) simultaneously for 1h. 1M NaOH was added dropwise to neutralize L-lysine and accelerate secondary chain extension. WPU without GQAS (WPU0) was prepared as a control with a similar process. WPUn films were prepared with a similar process to our previous work [33] before testing, including antibacterial, anti-biofilm activity and cytotoxicity.”
has been revised to
“2.2 Preparation of WPUn emulsions and coatings
2.2. Preparation WPUn emulsions and coatings
A series of self-polishing WPUn emulsions based on IPDI, PCL, PEG, lysine and EG12 were synthesized by two-step polymerization. The feed ratios were shown in Table 1. The content of GQAS in WPUn structure was regulated by changing the ratio of two chain extenders. PEG and PCL were dehydrated under vacuum at 80-90 ºC for 2 h first. Then IPDI and 0.1% stannous octoate were added under N2 environment. After continuous reaction for another 2 h at 70-75 ℃, the reaction prepolymer was cooled down. Then different content of chain extender EG12 in acetone was added in the mixture for 0.5 h. Afterward, the polyurethane prepolymer was poured into L-lysine aqueous solution under stirring (600 rpm) and ultrasound (200W) simultaneously for 1h. 1M NaOH was added dropwise to neutralize L-lysine and accelerate the secondary chain extension. WPU0 without GQAS was prepared as a control with a similar process.
WPUn coatings were prepared with a similar process to our previous work [33], by casting these WPUn emulsions siliconized culture dishes and dried at room temperature for 2 d, 60 ºC for 2 d, and then 60 ºC under vacuum for 2 d. The films were cut in sheets (1×1 cm, 0.5 mm thickness), and used directly for tests unless mentioned other-wise. To eliminate the influences of unreacted PEG and GQAS moieties onto WPUn films of their anti-fouling activities, all the films were immersed in water (110 rpm, 37 ºC) for 6h, then put into an oven at 60 ºC for 2 d, before testing, including antibacterial, anti-biofilm activity, protein adsorption assay, and cell cytotoxicity.” (Line 140-159)
- the description of the degradation is difficult to understand
Response: The manuscript has been revised accordingly. The sentence
“Weight loss of these WPUn films (10×10 mm in size, with approximately 0.5 mm thickness) were tested in PBS (pH = 7.4), and PBS + lipase PS (0.3 mg·mL-1) with a same process to our previous work [42, 43]. The weight loss was calculated as:
Where m0 represents the initial weight of WPUn films, and mt the dry weight after degradation for different time. The morphology of WPUn after degradation was characterized by a scanning electron microscope microscopy (SEM).”
has been revised to
“Hydrolytic degradation medium (PBS, pH = 7.4) and enzymatic degradation medium (PBS + lipase PS, pH = 7.4, lipase PS concentration: 0.3 mg·mL-1) were prepared before the test. WPUn films (1×1 cm, 0.5 mm thickness) were weighed (W0) and immersed in a glass tube containing 2mL degradation medium, then the vials were incubated with shaking (110rpm) at 37 ℃[42, 43]. Periodically removed samples were washed with distilled water three times, dried in vacuum oven at 25℃ to a constant weight (Wt). The degradation medium was changed every 2 d with fresh medium. The weight loss was calculated as:
The morphology of WPUn after degradation was characterized by a scanning elec-tron microscope microscopy (SEM).” (Line 163-172)
5) please clearly state how the experiments were done - in emulsions or did you prepare films from the emulsions (how?)
Response: The manuscript has been revised accordingly. The sentence
“WPUn films were prepared with a similar process to our previous work [32] before testing, including antibacterial, anti-biofilm activity and cytotoxicity.”
has been revised to
“WPUn coatings were prepared with a similar process to our previous work [33], by casting these WPUn emulsions siliconized culture dishes and dried at room temperature for 2 d, 60 ºC for 2 d, and then 60 ºC under vacuum for 2 d. The films were cut in sheets (1×1 cm, 0.5 mm thickness), and used directly for tests unless mentioned otherwise. To eliminate the influences of unreacted PEG and GQAS moieties onto WPUn films of their anti-fouling activities, all the films were immersed in water (110 rpm, 37 ºC) for 6h, then put into an oven at 60 ºC for 2 d, before testing, including antibacterial, anti-biofilm activity, protein adsorption assay, and cell cytotoxicity.” (Line 152-159)
Results and discussion
- line 222 and others please avoid eco-friendly where it is not fully proven
Response: We thank the reviewer for the critical comments and helpful suggestions. The word “eco-friendly” has been removed in the manuscript.
- how the emulsions were studied - did you dilute them? What about the particle size distribution?
Response: The manuscript has been revised accordingly. The sentence
“Particle sizer and zeta potential of WPUn emulsions were tested using a Zetasizer Nano ZS dynamic light-scattering (DLS) instrument (Malvern) at 25 ºC at an angle of 90°.”
has been revised to
“Particle size and zeta potential of WPUn emulsions (diluted with distilled water to about 0.02 wt % before the test) were tested using a Zetasizer Nano ZS dynamic light-scattering (DLS) instrument (Malvern) at room temperature at an angle of 90°.” (Line 237-239)
The particle size distribution of these samples added in the Supplementary information as follows.
Figure S1. Z-average size of WPUn emulsions.
- did you verify what was the products of the degradation?
Response: We thank the reviewer for the critical comments and helpful suggestions. Unfortunately, the products of the degradation were not verified in this work. Further structural analysis of degradation products is needed and to be studied in the future work.
- figure 3 please magnify the scale bar
Response: The manuscript has been revised accordingly. Scale bar has been added in caption of Figure 3 and 5 accordingly.
The caption of Figure 3
“Surface morphology of WPUn. a0-d0: WPUn non-degraded; a1-d1: WPUn degraded in PBS; a2-d2: WPUn degraded in PBS + lipase PS.”
has been revised to read
“Surface morphology of WPUn. a0-d0: WPUn non-degraded; a1-d1: WPUn degraded in PBS; a2-d2: WPUn degraded in PBS + lipase PS. Scale bar = 10 μm.” (Line 305-306).
The caption of Figure 5
“Morphology of E. coli attached to WPUn films (10000 ×) after 2 days incubation in E. coli culture (107CFU·mL-1). (a) LB media; (b) LB+0.5% casamino acids media”
has been revised to read
“Morphology of E. coli attached to WPUn films (10000 ×) after 2 days incubation in E. coli culture (107CFU·mL-1). (a) LB media; (b) LB+0.5% casamino acids media. Scale bar = 10 μm.” (Line 415-416).
- section 3.3.4. how did you verify no bacteria, even from the air?
Response: We thank the reviewer for the critical comments and helpful suggestions. The contact-active antibacterial activity test was performed according to the following process.
“50μl WPUn emulsions were casted onto cover glasses (1.5×1.5 cm2) and dried completely. Then all these samples were immersed in water at 37 ºC for 6h and dried to exclude the effects of small molecular antibacterial components. After 2h UV-sterilization, these samples were sprayed with an aqueous suspension of S. aureus (1×106 cells·mL-1) and air-dried for 10 min, then incubated in nutrient broth agar (0.8% agar) media at 37 ºC for 16-24 h. Bacterial colonies grew from the individual cells and stained red with 3 mL triphenyltetrazolium chloride (5mg·mL-1).”
The pictures of contact-active antibacterial activity test have been changed backed to original image (Figure 4c), that these red dots are bacterial colonies grew from the individual cells after stained red with 3 mL triphenyltetrazolium chloride. Square area was coated by WPUn emulsions, that no red dots meaning no bacteria. (Line 362)
Figure 4. Anti-fouling activity of WPUn films. (a) Time-related water contact angle of WPUn films; (b) Nonspecific protein adsorption of WPUn films; (c) Contact-active antibacterial activity of WPUn films; (d) Number of living bacteria attached on WPUn under different bacterial culture (E. coli-LB+: LB+0.5%vcasamino acids; E. coli-LB: LB; S. aureus-TSB+: (TSB+2% glucose; S. aureus-TSB: TSB).
- why there are no curves from DSC and TGA in the manuscript?
Response: We thank the reviewer for the critical comments and helpful suggestions. TGA test was not performed and discussed in this manuscript. The DCS curves are illustrated in the supplementary material as Figure S4.
Figure S4. DSC thermograms of WPUn (second heating).

Round 2
Reviewer 1 Report
Authors improved manuscript significantly according to expected revision. The present form of manuscript can be accepted for publication.
Author Response
Authors improved manuscript significantly according to expected revision. The present form of manuscript can be accepted for publication.
Response: Thank you for your recognition!
Reviewer 2 Report
The Authors partially improved the manuscript. Still, major revision is needed. Detailed comments were listed below:
Abstract
- updated - thank you
Introduction
- line 101 cross out "ideal"
Results and discussion
- still, there are no DSC curves in the main body text. Furthermore, why did you perform DSC only till 100 deg? Why there are only curves after the second heating?
- z-ave is not the same as the presented results in supporting materials. Please have look a at this paper https://doi.org/10.3390/molecules26195856
where presented how the results of DLS should be discussed and what is the differences between intensity and number
The data such as FTIR and DSC should be inserted in the main body text
Round 3
Reviewer 2 Report
The Authors improved the manuscript due to my comments.
Please keep in mind that dilution "kills" the emulsion system. I m understand the limitation of the apparatus which measures 90 deg. For that reason, DLS can be considered here only for comparing the behavior of the system. The Author may add this statement in the results and discussion section, and information that further studies (if necessary) might be focused on detailed emulsion characterization as presented i.e. here https://doi.org/10.3390/molecules26195856
Author Response
Q1. The Authors improved the manuscript due to my comments. Please keep in mind that dilution "kills" the emulsion system. I m understand the limitation of the apparatus which measures 90 deg. For that reason, DLS can be considered here only for comparing the behavior of the system. The Author may add this statement in the results and discussion section, and information that further studies (if necessary) might be focused on detailed emulsion characterization as presented i.e. here https://doi.org/10.3390/molecules26195856
Response: Thank you for your recognition! The statement has been added in the results and discussion section accordingly.
The sentence
“Zeta potential of these WPUn emulsions is shown in Table 1”
has been revised to
“Since dilution may affect the emulsion stability [49], zeta potential of these WPUn emulsions (Table 1) obtaining from DLS is only for comparing the behavior of this system.” (Line 251-253)
[49] W. Smułek, P. Siejak, F. Fathordoobady, Ł. Masewicz, Y. Guo, M. Jarzębska, D.D. Kitts, P.Ł. Kowalczewski, H.M. Baranowska, J. Stangierski, A. Szwajca, A. Pratap-Singh, M. Jarzębski, Whey Proteins as a Potential Co-Surfactant with Aesculus hippocastanum L. as a Stabilizer in Nanoemulsions Derived from Hempseed Oil, Molecules 26(19) (2021) 5856.